# Different Tea Germplasms Distinctly Influence the Adaptability of *Toxoptera aurantii* (Hemiptera: Aphididae)

**DOI:** 10.3390/insects14080695

**Published:** 2023-08-07

**Authors:** Changhao Lu, Ni Shen, Wenbin Jiang, Bi Xie, Runa Zhao, Guolan Zhou, Degang Zhao, Yingqin He, Wenlong Chen

**Affiliations:** 1Guizhou Provincial Key Laboratory for Agricultural Pest Management of the Mountainous Region, Institute of Entomology, Scientific Observing and Experimental Station of Crop Pest in Guiyang, Ministry of Agriculture, Guizhou University, Guiyang 550025, China; lchao10@163.com (C.L.); zhao21373@163.com (R.Z.); 2College of Tea Science, Guizhou University, Guiyang 550025, China; jiangwb990825@163.com (W.J.); 18485787180@163.com (B.X.); 3Guizhou Plant Conservation Center, Guizhou Academy of Agriculture Science, Guiyang 550006, China; 17508516139@163.com (N.S.); ckszgl@163.com (G.Z.); dgzhao@gzu.edu.cn (D.Z.)

**Keywords:** *Toxoptera aurantii*, electropenetrography, feeding behavior, life table, host adaptability

## Abstract

**Simple Summary:**

The tea aphid, *Toxoptera aurantii*, is one of the most damaging pests of tea plants and substantially affects tea yield. Thus, understanding *T. aurantii*’s adaptability to various tea germplasms is critical for screening resistant materials. Here, we used EPG technology combined with an age-stage two-sex life table to assess *T. aurantii*’s adaptation to six tea germplasms. The findings revealed that various tea germplasms substantially impacted *T. aurantii* feeding behavior. Compared with other hosts, *T. aurantii* exhibited increased pathway activities and reduced phloem sap ingestion on ZK. Life table parameters indicated that the growth rate of *T. aurantii* was the slowest on ZK, and it also significantly affected the longevity and fecundity of *T. aurantii*. Thus, ZK was the least favorable host plant for *T. aurantii* population establishment among the six germplasms based on measurements of feeding behavior and population dynamics.

**Abstract:**

Aphids are typical phloem-sucking insect pests. A good understanding regarding their feeding behavior and population dynamics are critical for evaluating host adaptation and screening of aphid-resistant resources. Herein, the adaptability of *Toxoptera aurantii* (Boyer) (Hemiptera: Aphididae) to different hosts was evaluated via electropenetrography and an age-stage, two-sex life table on six tea germplasms: Zikui (ZK), Zhongcha108 (ZC108), Zhongcha111 (ZC111), Qianmei419 (QM419), Meitan5 (MT5), and Fudingdabaicha (FD). Our findings revealed that the feeding activities of *T. aurantii* differed considerably among the host plants. *T. aurantii* exhibited significantly more pathway activities on ZK and FD than on the other hosts. However, the duration of feeding of *T. aurantii* on ZK phloem considerably decreased compared with those of the other germplasms. Life parameters indicated that *T. aurantii* exhibited the highest intrinsic rate of increase (*r*), net reproductive rate (*R*_0_), and finite rate of increase (*λ*) on MT5, and the maximum values of total longevity and oviposition period were recorded on FD; these variables were reduced significantly on ZK. The results of our study demonstrate that *T. aurantii* can successfully survive on the six tea germplasms; however, ZK was less suitable for *T. aurantii* and should be considered as a potential source of resistance in breeding and Integrated Pest Management.

## 1. Introduction

Via the interactions between phytophagous insects and host plants, the former has evolved adaptability to host plants in complex ecological environments [1]. Through this process, insects select plants as food sources through a range of behavioral and physiological responses to improve their adaptability to host plants, thus leading to the diversity or specialization of various insects to a specific host, and ultimately making insect populations more rampant on some crops [2]. Increasing evidence suggests that the performance of insects on host plants is widely considered a reliable indicator for evaluating their adaptability [3,4,5]. Therefore, studying insects’ feeding behavior and population dynamics, we can judge their adaptability to various host plants, which is also essential for developing insect-resistant host plant varieties [6].

The tea aphid, *Toxoptera aurantii* (Boyer) (Hemiptera: Aphididae), is one of the most serious pests of tea plants, which damages them by sucking sap from the phloem and producing metabolites such as honeydew to increase the risk of sooty mold [7]. Reportedly, *T. aurantii* causes an average yield loss of 5−55% annually [8]. Nowadays, *T. aurantii* is widely distributed in countries such as China, India, Japan, Sri Lanka, and Kenya, it also can harm other plants such as cocoa, coffee, and citrus [9]. Given the high adaptability of *T. aurantii*, they can further increase their pest potential in the near future. The current methods of managing aphids focus mostly on insecticides [10,11]. In attempting to avoid the negative effects of chemical insecticides, the utilization of resistant tea germplasms to control aphids has emerged as an economical, effective, and environmentally friendly management strategy [12,13]. Unfortunately, few reports regarding the resistance of tea plants to *T. aurantii* have been published, which are insufficient for screening aphid-resistant tea germplasms.

When aphids encounter a new host plant, they probe the plant tissues to determine the resistance of the epidermis and appropriateness of the phloem sap, and only accept the plant as a food source following this assessment procedure. In order to complete their life cycle, aphids also assess whether the host plant can provide a suitable habitat for their development and reproduction. Therefore, combining knowledge of their feeding behavior with population dynamics could offer a better understanding regarding aphid adaptation to diverse host plants [14]. Owing to their efficiency and accuracy, the electrical penetration graph (EPG) technique and life tables are useful tools for screening insect-resistant germplasm resources to comprehensively analyze the adaptation of piercing–sucking insects to plants [12]. EPG signals can reflect the location of the stylet in the plant tissue alongside special feeding activities [15,16,17] and has been widely used as an advanced technology to examine the feeding behavior of piercing–sucking insects [18] such as *Rhopalosiphum padi* (Linnaeus) (Hemiptera: Aphididae) [19], *Toxoptera citricida* (Kirkaldy) (Hemiptera: Aphididae) [13], *Myzus persicae* (Sulzer) (Hemiptera: Aphididae) [20], and *Bemisia tabaci* (Gennadius) (Hemiptera: Aphididae) [21]. Life table analysis is widely accepted as a powerful tool for researching the dynamics of insect populations. The intrinsic rate of increase (*r*), net reproductive rate (*R*_0_), and finite rate of increase (*λ*) obtained from the age-stage, two-sex life table analysis can accurately reveal the dynamics of insect populations and provide accurate findings for evaluating host adaptability [20].

We recorded the feeding behavior of *T. aurantii* on six tea germplasm materials using the EPG technique and assessed the growth and development levels of aphids in relation to the life table parameters in the current study. The overarching goal of this study was to clarify the host adaptability of *T. aurantii* and filter out resistant tea germplasm sources to establish a theoretical foundation for prospective pest preventive and control actions during tea planting.

## 2. Materials and Methods

### 2.1. Test Materials

#### 2.1.1. Insects

*T. aurantii* were collected from the tea field in the teaching experimental farm of Guizhou University. These aphids were reared in an insect-proof cage in a greenhouse at (25 ± 1) °C, (70 ± 5)% RH, and 14 L:10 D photoperiod. In order to avoid the interference of other germplasm materials on the adaptability of *T. aurantia.* Before the experiment, an apterous adult from the stock colonies was transferred to a susceptible tea seedling *Camellia sinensis* cv. Huangjinya. After three generations of breeding, the newly emerged (0–6 h) adults of *T. aurantii* were picked out for the studies.

#### 2.1.2. Tea Germplasm Materials

Tea germplasm materials Zikui (ZK), Zhongcha108 (ZC108), Zhongcha111 (ZC111), Qianmei419 (QM419), Meitan5 (MT5), and Fudingdabaicha (FD) were provided by the Institute of Tea Research, Guizhou Academy of Agricultural Sciences. All tea germplasms were 2-year-old tea seedlings grown in an artificial climate chamber (25 °C ± 1 °C, 70% ± 5% RH, 14 L:10 D photoperiod).

### 2.2. Test Methods

#### 2.2.1. EPG Recording

In this experiment, we used a DC-EPG Giga-8 system (EPG Systems, Wageningen Agricultural University, The Netherlands) to record the feeding behavior of *T. aurantii* on six germplasms. The newly molted adult apterous aphids were first starved for 1 h before binding the pronotum of *T. aurantii* to a thin gold wire (1~2 cm long and 12.5 µm in diameter) with silver conductive paint glue. The other end of the gold wire was fixed on the welded fine copper nail, and the nail was fixed to the probe of the input electrode to form a biotic electrode. Then, another copper rod with a diameter of 2 mm and a length of 10 cm was inserted into the soil of the plant to be measured to form the plant electrode. Finally, in accordance with the feeding habits of *T. aurantii*, it was placed on the abaxial surface of the upper leaf of the test plant to feed freely. When the aphid on the plant’s leaf pierced the plant tissue, it formed a closed circuit. The experiment was conducted in a Faraday cage to eliminate interference from electrical noise. The feeding behavior was continuously recorded for 6 h, following which we selected 15 valid duplicates for statistical analysis.

#### 2.2.2. Life Table Construction

The experiment was performed on live potted plants. The potted tea plants were placed in trays filled with water to prevent aphids from escaping and to ensure normal plant growth. An apterous adult was obtained from each of the six tea germplasms and placed on the appropriate fresh leaves of the tea plant. Every 12 h, the survival, molting, and number of newly born nymphs for each aphid were recorded, and exuviae and new aphids were promptly removed. Only one new aphid was permitted to be placed on each plant, and each tea germplasm was replicated 50 times. The experiment was carried out in a climate chamber at 25 °C ± 1 °C, 70% ± 5% RH, with a photoperiod of 14 L:10 D.

### 2.3. Statistical Analysis

The aphids’ probing behaviors were recorded using the Stylet+d program. The labeling of aphid waveforms on Stylet+a software according to Tjallingii [22,23,24] and Helden and Tjallingii [25] descriptions and all behavioral variables were processed using the EPG Excel Data Workbook produced by Sarria et al. [26]. The EPG parameter analyses were performed using SPSS 26.0 statistical software (SPSS, Chicago, IL, USA). Kolmogorov–Smirnov’s test and Bartlett’s test were used to test the normality and homogeneity of the variance of the data, respectively. In the absence of normality, the data of the EPG were log10(x + 1) transformed and arcsine conversion was used for data on the percentage of the average duration of each waveform. One-way ANOVA with Tukey–HSD test (*p* < 0.05) was conducted to detect differences between the treatments, with germplasms as the independent variable and EPG parameters as the dependent variable. Each tea germplasm was performed in 15 replicates. We assessed the growth and development of *T. aurantii* on six tea germplasm resources with 50 replicates per treatment.

The life table and particular age-related parameters were exported using a bootstrap approach with 100,000 resampled data values in the software TWOSEX-MSChart [27,28]. We used the above statistical method of EPG data to test the differences among the parameters. The *s_xj_*, *l_x_*, *m_x_*, *l_x_m_x_*, *e_xj_*, and *v_xj_* curves were drawn using SigmaPlot 14.

## 3. Results

### 3.1. Measuring T. aurantii Feeding Behavior by EPG

We observed seven distinct waveforms (np: non-probing behavior, pd: intracellular stylet puncture, C: intercellular stylet pathway, G: xylem sap ingestion, F: derailed stylet mechanics, E1: phloem salivation, and E2: passive phloem ingestion) on the six tea germplasms during the 6 h recording. The comparison of phloem phase and nonphloem phase parameters revealed that the feeding behavior of *T. aurantii* on ZK was significantly affected.

### 3.2. Probing Behavior of T. aurantii Stylets in Non-Phloem Phase

No significant differences were observed in the number of probes, number of probes to the 1st E1, number of short probes (C < 3 min), and total duration of F among six host plants (Table 1, Parameters 1, 4, 7, and 11). The total time of the np on MT5 was the longest among all the tea germplasm and even considerably longer than ZC108 (Table 1, Parameter 2). The duration of first probe was significantly extended on FD compared with ZC108 (Table 1, Parameter 3). *T. aurantii* exhibited the highest count of pd on ZK; however, the mean duration of pd increased significantly on MT5 (Table 1, Parameters 5 and 6). The total duration of C and the number of F showed the maximum value on ZK (Table 1, Parameters 8 and 10). The duration of *T. aurantii* on ZC111 and QM419 was significantly longer than that of other germplasms (Table 1, Parameter 9).

### 3.3. Probing Behavior of T. aurantii Stylets in Phloem Phase

There are no significant difference between *T. aurantii* on the six treatments in terms of the number of E1, number of E2, and number of sustained E2 (>10 min) (Table 1, Parameters 12, 14, and 15). The total duration of E1 on ZK was the highest among six tea germplasms (Table 1, Parameter 13). Conversely, the total duration of E2 was significantly lower on ZK, and the feeding behavior of *T. aurantii* was significantly affected as it spent less time pricking the phloem on ZK (Table 1, Parameter 16).

### 3.4. Average Percentage of Time per Waveform during 6 h

The distribution (as a percentage of the total) of the average durations of the various waveforms of *T. aurantii* on the six tea germplasms was analyzed (Figure 1). In the nonphloem phase, ZK attained the maximum value of 58.55% in C wave, and ZC108 reached the lowest value of 21.12%. ZC111 and QM419 were obviously longer than others for G wave. There was no difference in the proportion of F wave among the six tea germplasms. In the phloem phase, the proportion of E1 wave was the highest in ZK, which were consistent with parameter 13 shown in Table 1. The proportion of E2 waves exhibited highly significant differences, with ZC108 having the highest value at 69.29%, while ZK had the lowest value at only 20.59%.

### 3.5. Life Table Analysis

Population dynamics data (nymphal, preadult, and adult stages; fecundity; and total longevity) of *T. aurantii* on ZK, ZC108, ZC111, QM419, MT5, and FD tea germplasms are shown in Table 2. The developmental times of the preadult stage differed markedly among the six tea germplasm materials: *T. aurantii* developed the slowest on ZK and the quickest on MT5 during the early adult stage. Moreover, there were substantial differences (*p* < 0.01) in the time required for *T. aurantii* growth and development at each age among the six germplasm sources. During the adult stage, or even throughout the whole growth and development stage, *T. aurantii* exhibited the shortest longevity on ZK. In addition, feeding on different tea germplasms induced changes in the APOP and TPOP, oviposition time, and fecundity. The APOP of *T. aurantii* reared on QM419 and ZK was significantly greater than that of the other four host plants. The TPOP values measured on ZK were considerably longer than those recorded on others. The longest oviposition period and highest fecundity occurred on FD and MT5, respectively, whereas the minimum values of both the variables were observed on ZK. Among the six germplasms, there was no difference in the survival rate of *T. aurantii* in the larval stage, with all aphids surviving to the adult stage.

Table 3 shows the parameters of the population that were estimated using the bootstrap method. The highest values of the finite rate of increase (*λ*), intrinsic rate of increase (*r*), and reproductive rate (*R*_0_) appeared on MT5, whereas the lowest values occurred on ZK. Furthermore, the maximum and minimum values of the mean generation time (*T*) were detected on ZK and ZC108, respectively.

The age-specific survival rates (*l_x_*), age-specific fecundity values (*m_x_*), and net maternity values (*l_x_m_x_*) of *T. aurantii* on the six germplasm materials are shown in Figure 2. The findings revealed that the *l_x_* value gradually decreased with increasing age. The age-specific fecundity on ZK, ZC108, ZC111, QM419, MT5, and FD was 5.02, 4.16, 2.99, 3.30, 4.57, and 3.62, respectively. Figure 3 shows the probability that an individual neonate aphid will survive to age *x* and stage *j*. The overlap of survival curves at specific stages may be due to differences in developmental rates between individuals. The age stage-specific lifespan revealed the expected lifespan of individual *T. aurantii* of age *x* and stage *j* on the different tea germplasm materials (Figure 4). The life expectancy of a neonate was 18.04, 18.32, 18.99, 19.28, 19.02, and 20.13 days on ZK, ZC108, ZC111, QM419, MT5, and FD, respectively. The age stage-specific reproductive values (*v_xj_*) of *T. aurantii* on the different tea germplasm materials are shown in Figure 5. *T. aurantii* reared on six tea germplasms had the highest reproductive values in the early stages of maturity. The highest value for reproduction peaked at 14.02 on MT5, while the lowest peak occurred on ZK at 10.36, both of which occurred late on the sixth day.

## 4. Discussion

The host adaptability of aphids is closely related to the plant species [29,30]. To design ecologically effective pest control strategies, there is a need for a comprehensive understanding of aphid performance on different tea germplasms. Herein, the feeding behavior and population dynamics of *T. aurantii* on six tea germplasms were evaluated using the EPG technique and life table method.

The feeding habits of piercing–sucking insects can reflect a lot of information, which is useful in determining the aphids’ preference for host plants. After determination and analysis, *T. aurantii* produced seven waveforms (np, pd, C, G, F, E1, and E2) on the six tea germplasms, which was consistent with the conclusion of Han et al. [31]. When aphids feed on a plant, the stylets penetrate the epidermis and then suck its internal composition of the tissue [32]. Thus, the factors affecting the feeding of herbivorous insects include the physical and chemical properties of the plant surface and internal tissues [33]. The total duration of non-probing and pathway of *T. aurantii* on ZK was relatively high on the six tea germplasms; hence, we speculated that it is related to the physical characteristics of ZK leaves, such as their color and surface structure. The physical traits of plants exhibit a significant impact on aphid feeding activities and are often used in research on host adaptation. Hao et al. [34] reported that the feeding behavior of *Brevicoryne brassicae* (L.) was negatively correlated with the length and density of trichomes. Similar results were confirmed for *M. persicae* and *Melanaphis sacchari* (Zehntner) [35,36].

The phloem sap of host plants constitutes the main nutrient for aphids. E1 and E2 waveforms are indicative of phloem feeding activity that may be utilized to evaluate aphid preferences on host plants [37]. The E1 wave indicates secretion of water-soluble saliva by the stylets after reaching the sieve tube [38,39]. The total duration of E1 on ZK was the longest of any test material, it may suggest that *T. aurantii* must produce a large amount of saliva to overcome associated defense substances in order to feed successfully from the plant phloem. E2 is the waveform that represents passive feeding on sap in the phloem. The number of sustained E2 waves (>10 min) on MT5 was higher than that on ZK in the current study. In addition, the duration of E2 was the lowest on ZK and highest on ZC108. Liang et al. [40] reported that the total duration of E2 of *T. aurantii* on ZC108 was significantly reduced. This result is inconsistent with the conclusion of our study, which might be due to a regional difference in ZC108.

The life table parameters of herbivorous insects can also reflect their adaptability to different host plants. The population parameters were produced using the bootstrap method with 100,000 resamplings. The results revealed that feeding on different tea germplasms can change the biological parameters of *T. aurantii*. Increasing evidence suggests that a short development time, extended longevity, and the large reproductive capacity of phytophagous insects are features that are essential for a high capacity to adapt to different hosts [41,42,43]. In this study, the nymph period of *T. aurantii* was clearly longer on ZK than others, and its fecundity was significantly reduced. Certain biochemical or physical disorders may reduce *T. aurantii’s* feeding on ZK; hence, lowering its developmental and reproductive capacity. Undoubtedly, this will be a part of our further research.

The population parameters (*r*, *λ*, and *R*_0_) reflect the physiological characteristics related to fertility of the insects [44,45]. Among the six tea germplasm materials, the maximum and minimum values of these three variables occurred on MT5 and ZK, respectively. Tea plant resistance to insect pests is the consequence of a complex interplay of factors, including leaf bud color, physical characteristics, biochemical composition, and tenderness-keeping ability, etc. Yang et al. [46] found that ZC108 and FD had low caffeine content but high free amino acid content, and that *Dendrothrips minowai* (Priesner) and *Empoasca onukii* (Matsuda) were more adaptable to these two host plants. According to Luo et al. [47], there is a significant positive correlation between plant anthocyanin content and insect resistance, and the greater the anthocyanin content in leaves, the better the resistance against *Apolygus lucorm* (Meyer-Dür). Moreover, plants with strong tenderness-keeping ability can prolong aphid longevity and larviposition period, thus improving their fecundity. Overall, *T. aurantii* was the most adaptable on MT5, whereas the opposite was observed on ZK. MT5 is an excellent strain previously selected by our research group that has strong tenderness-keeping ability. Previous results demonstrated that ZK exhibited a high content of anthocyanins and short tender-maintaining time [48]. However, other resistance factors for ZK require further study.

In conclusion, the results measure adaptation to each of the tested tea germplasm types, and show that *T. aurantii* is better adapted to MT5 than others, whereas ZK was the least suitable host plant. Therefore, ZK can be regarded as a potential aphid-resistant germplasm for further study to form a useful genetic resource.

## Figures and Tables

**Figure 1 insects-14-00695-f001:**
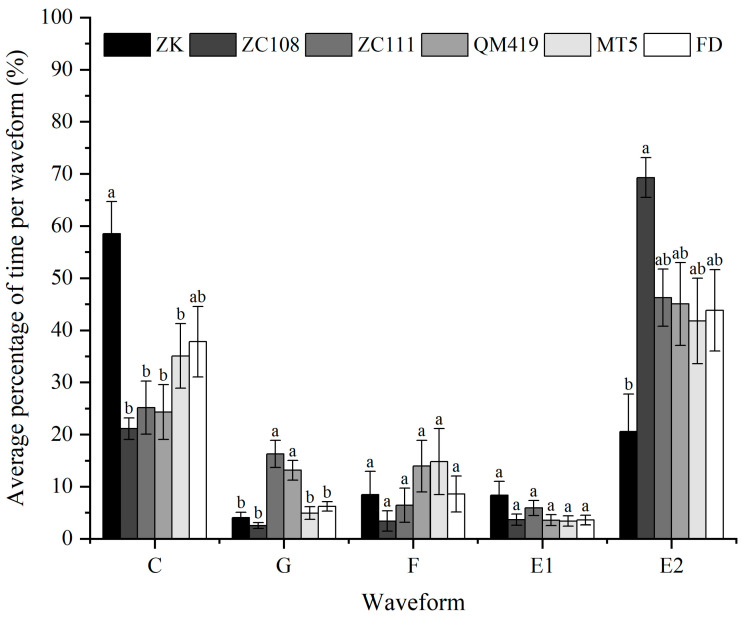
Percentage of time spent by *T. aurantii* (*n* = 15 per treatment) in each waveform throughout each 6 h recording on six tea germplasms. Significant differences are shown by different letters (Tukey–HSD test, *p* < 0.05). Pathway waveforms (C), xylem activities (G), derailed stylet mechanics (F), phloem salivation (E1), and phloem ingestion (E2).

**Figure 2 insects-14-00695-f002:**
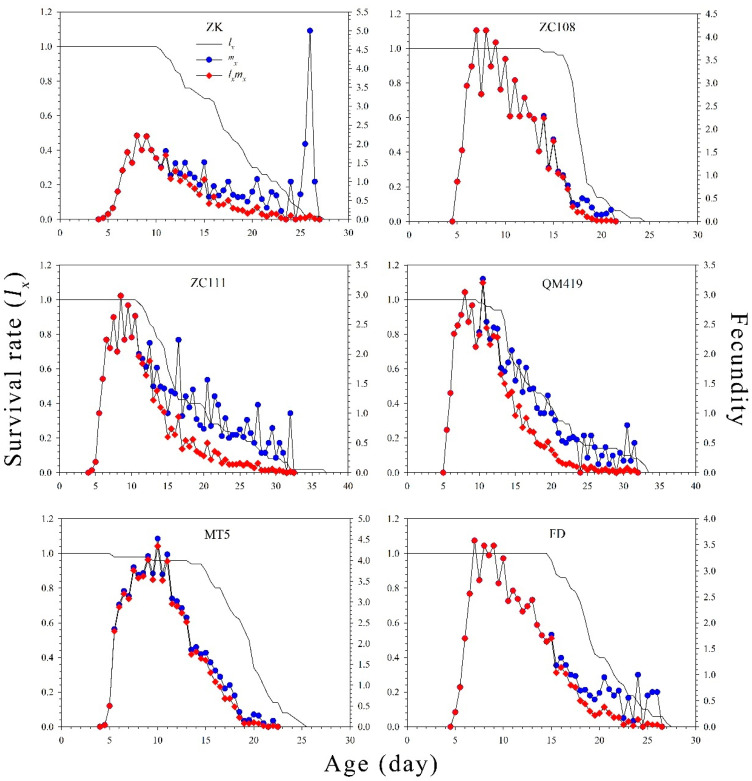
Age-specific survival rate (*l_x_*), age-specific fecundity of the total population (*m_x_*), and net maternity (*l_x_m_x_*) of *T. aurantii* on six different tea germplasm materials.

**Figure 3 insects-14-00695-f003:**
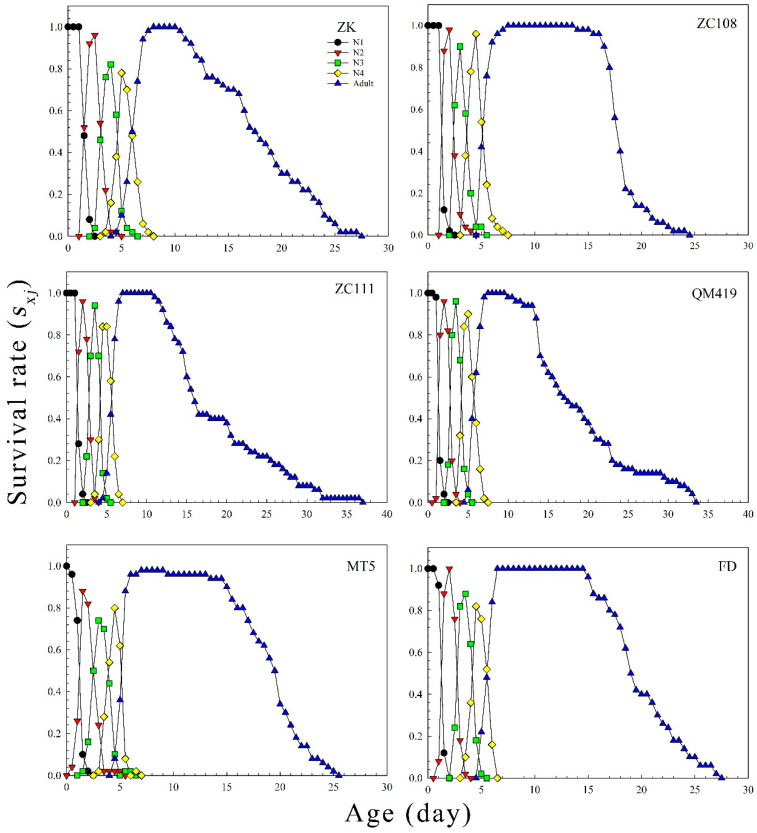
*T. aurantii* age stage-specific survival rate (*s_xj_*) on six distinct tea germplasm sources. N1 is the first instar nymph, N2 is the second instar nymph, N3 is the third instar nymph, and N4 is the fourth instar nymph.

**Figure 4 insects-14-00695-f004:**
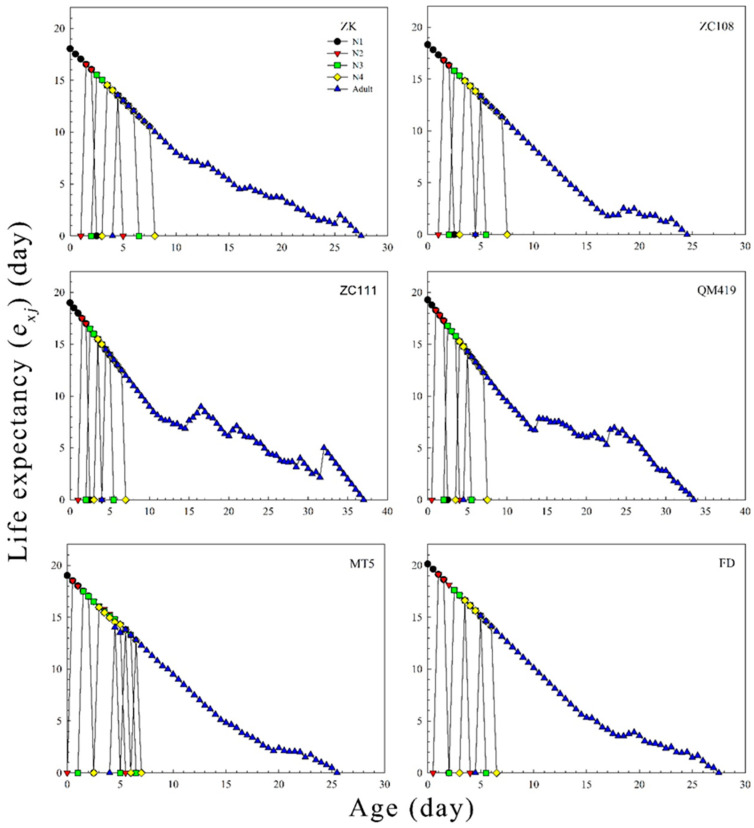
The age-stage life expectancy (*e_xj_*) of *T. aurantii* on six different tea germplasm materials. N1, first instar nymph; N2, second instar nymph; N3, third instar nymph; N4, fourth instar nymph.

**Figure 5 insects-14-00695-f005:**
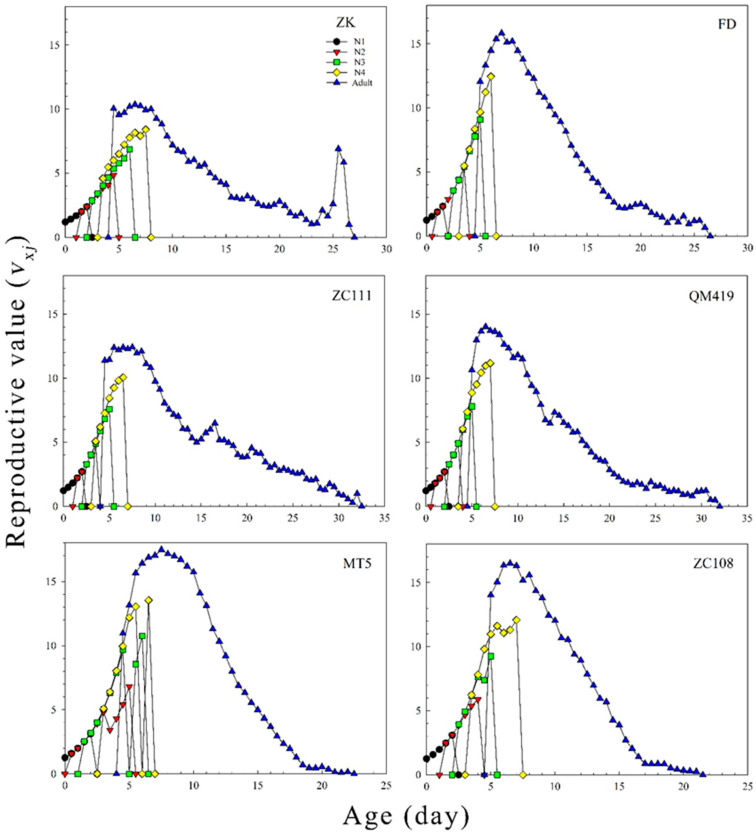
Age-stage reproductive value (*v_xj_*) of *T. aurantii* on six different tea germplasm materials. N1, first instar nymph; N2, second instar nymph; N3, third instar nymph; N4, fourth instar nymph.

**Table 1 insects-14-00695-t001:** Electrical penetration graph parameters of *T. aurantii* on six tea germplasm materials.

No.	EPG Parameters	Tea Germplasms						*p*
		ZK	ZC108	ZC111	QM419	MT5	FD	
1	Count probes	14.60 ± 3.71	6.60 ± 1.77	14.60 ± 2.59	11.20 ± 2.39	11.67 ± 2.72	6.93 ± 3.18	0.171
2	Sum time of np wave (min)	20.33 ± 8.23 ab	2.76 ± 0.60 b	8.46 ± 2.22 ab	6.88 ± 1.89 ab	31.77 ± 13.34 a	7.13 ± 5.68 ab	0.035
3	Duration of 1st probe (min)	69.88±26.64 ab	38.88 ± 23.54 b	98.70 ± 37.59 ab	79.67 ± 30.97 ab	88.71 ± 38.62 ab	204.50 ± 44.75 a	0.026
4	Number of probes to the 1st E1	6.13 ± 1.81	5.40 ± 1.62	10.07 ± 1.99	5.33 ± 1.55	9.60 ± 2.82	6.87 ± 3.19	0.508
5	Count of Pd	122.40 ± 16.22	77.60 ± 7.51	112.47 ± 10.18	102.20 ± 14.15	92.00 ± 9.20	92.40 ± 12.26	0.122
6	Mean duration of pd (s)	4.25 ± 0.31 ab	4.05 ± 0.13 ab	4.02 ± 0.11 ab	3.82 ± 0.84 b	4.95 ± 0.47 a	4.07 ± 0.98 ab	0.033
7	Number of short probes	8.00 ± 2.71	3.73 ± 1.34	8.67 ± 1.94	6.00 ± 1.55	6.27 ± 2.08	4.53 ± 2.50	0.521
8	Total duration of C (min)	218.57 ± 25.59 a	77.78 ± 8.15 c	89.41 ± 17.48 c	111.72 ± 27.05 bc	110.54 ± 15.62 bc	186.45 ± 35.34 ab	<0.01
9	Total duration of G (min)	13.87 ± 3.63 b	9.09 ± 2.00 b	57.09 ± 9.08 a	46.21 ± 6.52 a	17.32 ± 4.29 b	21.78 ± 3.21 b	<0.01
10	Number of F	5.00 ± 0.97 a	2.07 ± 0.54 b	1.80 ± 0.73 b	1.73 ± 0.50 b	1.07 ± 0.56 b	1.93 ± 0.61 b	<0.01
11	Total duration of F (min)	24.13 ± 11.24	12.15 ± 7.01	22.60 ± 11.58	48.57 ± 17.00	51.03 ± 22.00	30.28 ± 12.47	0.343
12	Number of E1	2.73 ± 0.62	2.87 ± 0.54	3.27 ± 0.80	2.40 ± 0.58	3.20 ± 0.86	1.73 ± 0.33	0.564
13	Total duration of E1 (min)	28.76 ± 9.06	13.02 ± 3.84	20.77 ± 5.05	12.61 ± 3.62	11.63 ± 3.30	12.76 ± 3.32	0.125
14	Number of E2	1.33 ± 0.47	2.27 ± 0.36	2.53 ± 0.65	2.00 ± 0.54	2.67 ± 0.80	1.13 ± 0.20	0.244
15	Number of sustained E2 (>10 min)	0.53 ± 0.17	1.47 ± 0.22	1.47 ± 0.26	1.13 ± 0.22	1.67 ± 0.63	0.80 ± 0.11	0.094
16	Total duration of E2 (min)	73.17 ± 25.72 b	247.56 ± 13.71 a	162.93 ± 19.45 ab	159.91 ± 28.40 ab	141.31 ± 29.25 b	155.88 ± 28.33 ab	<0.01

The feeding behavior of *T. aurantii* on six tea germplasms was recorded with EPG, and seven waveforms were determined: np (non-probing behavior), pd (intracellular stylet puncture), C (intercellular stylet pathway), G (xylem sap ingestion), F (derailed stylet mechanics), E1 (phloem salivation), and E2 (passive phloem ingestion). Data in the table represent mean ± SE (*n* = 15). Different letters in the same rows show significant differences between treatments at *p* < 0.05 (Tukey–HSD test).

**Table 2 insects-14-00695-t002:** Developmental stage duration and growth parameters of *T. aurantii* on six tea germplasm materials.

Parameters	Tea Germplasms						*p*
	ZK	ZC108	ZC111	QM419	MT5	FD	
First Instar (d)	1.78 ± 0.05 a	1.57 ± 0.03 bc	1.66 ± 0.04 ab	1.61 ± 0.04 b	1.41 ± 0.05 c	1.52 ± 0.03 bc	<0.01
Second Instar (d)	1.60 ± 0.04 a	1.20 ± 0.04 c	1.39 ± 0.04 bc	1.42 ± 0.03 ab	1.41 ± 0.08 ab	1.46 ± 0.04 ab	<0.01
Third Instar (d)	1.42 ± 0.05 a	1.19 ± 0.03 b	1.36 ± 0.04 a	1.41 ± 0.03 a	1.35 ± 0.05 ab	1.39 ± 0.04 a	<0.01
Fourth Instar (d)	1.43 ± 0.06 ab	1.52 ± 0.04 ab	1.43 ± 0.06 ab	1.61 ± 0.08 a	1.17 ± 0.05 c	1.36 ± 0.04 bc	<0.01
Preadult Duration (d)	6.23 ± 0.11 a	5.48 ± 0.08 cd	5.84 ± 0.08 b	6.05 ± 0.08 ab	5.35 ± 0.07 d	5.73 ± 0.07 bc	<0.01
Adult Longevity (d)	11.77 ± 0.67	12.84 ± 0.27	13.15 ± 0.93	13.23 ± 0.89	13.66 ± 0.51	14.40 ± 0.47	0.13
Total Longevity (d)	18.00 ± 0.66	18.32 ± 0.26	18.99 ± 0.93	19.28 ± 0.90	19.01 ± 0.52	20.13 ± 0.47	0.28
APOP (d)	0.15 ± 0.01 a	0.02 ± 0.01 c	0.11 ± 0.01 b	0.16 ± 0.01 a	0.12 ± 0.01 b	0.11 ± 0.01 b	<0.01
TPOP (d)	6.38 ± 0.12 a	5.50 ± 0.08 e	5.94 ± 0.08 c	6.22 ± 0.10 b	5.47 ± 0.07 e	5.85 ± 0.07 d	<0.01
Larviposition Period (d)	8.34 ± 0.07 f	10.66 ± 0.02 c	10.14 ± 0.09 d	9.82 ± 0.08 e	11.05 ± 0.06 b	11.47 ± 0.05 a	<0.01
Fecundity	31.22 ± 2.11 d	58.48 ± 1.18 a	45.90 ± 2.82 c	49.16 ± 2.68 bc	66.04 ± 2.20 a	57.42 ± 1.89 ab	<0.01

Means ± SE within rows followed by the same letter do not differ significantly according to the Tukey–HSD test (*p* < 0.05). TPOP represents the total prereproductive period, and APOP represents the adult prereproductive period.

**Table 3 insects-14-00695-t003:** Life table parameters of *T. aurantii* on six germplasm materials.

Parameters	Tea Germplasms						*p*
	ZK	ZC108	ZC111	QM419	MT5	FD	
*λ* (d^−1^)	1.4147 ± 0.0143 e	1.5751 ± 0.0011 b	1.4890 ± 0.0013 d	1.4880 ± 0.0012 d	1.5831 ± 0.0011 a	1.5230 ± 0.0011 c	<0.01
*r* (d^−1^)	0.3469 ± 0.0010 e	0.4543 ± 0.0007 b	0.3981 ± 0.0009 d	0.3974 ± 0.0008 d	0.4594 ± 0.0007 a	0.4207 ± 0.0007 c	<0.01
*R* _0_	31.3552 ± 0.2849 e	58.3408 ± 0.1659 b	46.7204 ± 0.4117 d	49.3016 ± 0.3898 c	66.0652 ± 0.3344 a	57.2232 ± 0.2227 b	<0.01
*T* (days)	9.9289 ± 0.0237 a	8.9507 ± 0.0108 e	9.6523 ± 0.0232 c	9.8062 ± 0.0239 b	9.1216 ± 0.0125 d	9.6201 ± 0.0152 c	<0.01

Data in the table are represented as mean ± SE. Standard errors were estimated with 100,000 bootstrap resamplings. Means in each row followed by the same letter are not significantly different at *p* < 0.05 (Tukey–HSD test). Parameters of life table: Finite rate of increase (*λ*), Intrinsic rate of increase (*r*), Net reproductive rate (*R*_0_), and mean generation time (*T*).

## Data Availability

The data presented in this study are available on request from the corresponding author.

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
