# Peer review of "Different Tea Germplasms Distinctly Influence the Adaptability of Toxoptera aurantii (Hemiptera: Aphididae)"

_insects, 2023, doi:10.3390/insects14080695_

Round 1

Reviewer 1 Report

This study examines the adaptability of the aphid Toxoptera aurantii to several strains of tea by evaluating its feeding activities and life table characteristics on different tea strains to find the most resistant tea strain. This study thoroughly evaluated a number of aphid characteristics on several tea strains with sufficient sample size, and in this respect, I highly appreciate the authors’ endeavor. I agree with the authors conclusion on the most resistant strain of tea. However, the scope of the study is very limited, and I think that it may be difficult to attract the attention of many readers. On the other hand, the methods used by the authors may be useful for other studies to find resistant host strains for herbivorous insects.

My largest concern is related to the source of Toxoptera aurantia aphids used in the experiments. Why did not the authors use one or two clones for the research? Aphid clones have different adaptability to the host plant, with different life cycle characteristics. A clone may adapt to MT5, while a different clone may adapt to ZK. If the authors used aphids collected from MT5, they should obtain the results in this manuscript. So, in this kind of study, it is necessary to examine the characteristics of each aphid clone. Explain why it is possible to obtain a general conclusion by using aphids collected somewhere.

The authors used one-way ANOVA with Fisher’s Protected LSD test. However, Fisher's LSD is not appropriate because it does not correct for multiple comparisons. I recommend the authors to use Tukey’s multiple comparison procedure, which considers multiple comparisons. There are many explanations on this point. e.g.,https://peerj.com/articles/10387.pdf

Minor comments are,

Fig. 2. For ZK, mx values increased suddenly after day 25. Why is it? Is there any contamination?

Fig. 3. I do not think that the graphs represent “survival rates” of the aphid. This graph shows the proportion of each instar stages and adults when the initial number of the first instars is fixed as 1. For each instar stage, the curve first increases, levels off, and then decreases. Survival rates do not move like this.

Line 229 and 232. Figure. 2 and Figure. 3 should be Figure 2 and Figure 3.

Line 233. Remove “turnip”. 

Line 248. [282830] ??

Line 253-254. Not clear. Rephrase it.

Line 300. “oviposition” should be “larviposition”.

Quality of English seems to be high. No problem.

Author Response

Dear Editors and Reviewers:

Thank you for your letter and for the reviewers’ comments concerning our manuscript (ID: insects-2462631). Those comments are all valuable and very helpful for revising and improving our paper, as well as the important guiding significance to our researches. The manuscript has been carefully revised, and the revisions were marked in red. The point-by-point answers to the comments and suggestions were listed below.

Response to Reviewer 1 Comments

Point 1: My largest concern is related to the source of Toxoptera aurantii aphids used in the experiments. Why did not the authors use one or two clones for the research? Aphid clones have different adaptability to the host plant, with different life cycle characteristics. A clone may adapt to MT5, while a different clone may adapt to ZK. If the authors used aphids collected from MT5, they should obtain the results in this manuscript. So, in this kind of study, it is necessary to examine the characteristics of each aphid clone. Explain why it is possible to obtain a general conclusion by using aphids collected somewhere.

Response 1: We thank the reviewer for pointing out this issue. I did not elaborate on the source of Toxoptera aurantii in the article, so the reviewers were confused. T. aurantii was collected from the tea field in the teaching experimental farm of Guizhou University. Before the experiment, an apterous adult from the stock colonies was transferred to Huangjinya seedlings, and after three generations, the newly emerged (0-6h) adult of T aurantii were picked out for studies. The corrections list in line 94-100.

Point 2: The authors used one-way ANOVA with Fisher’s Protected LSD test. However, Fisher's LSD is not appropriate because it does not correct for multiple comparisons. I recommend the authors to use Tukey’s multiple comparison procedure, which considers multiple comparisons. There are many explanations on this point. e.g., https://peerj.com/articles/10387.pdf

Response 2: Thank you for your valuable advice. I carefully read the article you recommended and found Fisher’s Protected LSD test to be relatively liberal and do not make acceptable error rate adjustments. In addition, I looked through some articles similar to this issue and found that most people also use Tukey-HSD test. When making multiple comparisons, Tukey-HSD test is more reliable. So I used Tukey-HSD test to redo the multiple comparisons.

Point 3: Fig 2. For ZK, mx values increased suddenly after day 25. Why is it? Is there any contamination?

Response 3: Thank you for your questions. The preadult duration of aphids is different in different materials. The longer the preadult duration is, the slower the development of aphids is. As a result, the reproduction ability of aphids was affected, and the peak of aphid production appeared in the later stage of growth and development. In Table 2, it can be seen that the preadult duration of T. aurantii on ZK is significantly higher than that on other tea germplasm, and the mx curve in Figure 2 indicates the age-specific fecundity values, so the mx curve of T. aurantii on ZK will suddenly increase in the late stage. A similar pattern has emerged in other studies, here are the references:

  1. Yu, L. B; Lin, K.J; Xu, L.B; Wang, H; Cui, J; Zhang, Q.Y; Wang, Y.P; Yan, L.Y. Effect of different alfalfa cultivars on growth and development of the spotted alfalfa aphid, Therioaphis trifolii (Monell). Entomological Research. 2022, 52, 118-126. [ http://doi.org/10.1111/1748-5967.12571]
  2. Ali, M.Y.; Naseem, T.; Arshad, M.; Ashraf, I.; Rizwan, M.; Tahir, M.; Rizwan, M.; Sayed, S.; Ullah, M.I.; Khan, R.R.; Amir, M.B.; Pan, M.Z.; Liu, T.X. Host-plant variations affect the biotic potential, survival, and population projection of Myzus Persicae (Hemiptera: Aphididae). Insects. 2021, 1, 375-387. [https://doi.org/10.3390/insects12050375]
  3. Ali, G and Ebru, G. Influence of Different Hazelnut Cultivars on Some Demographic Characteristics of the Filbert Aphid (Hemiptera: Aphididae). Journal of Economic Entomology. 2017, 110, 1856-1862. [http://doi.org/10.1093/jee/tox087]

Point 4: Fig. 3. I do not think that the graphs represent “survival rates” of the aphid. This graph shows the proportion of each instar stages and adults when the initial number of the first instars is fixed as 1. For each instar stage, the curve first increases, levels off, and then decreases. Survival rates do not move like this.

Response 4: Figure 3 shows the age-stage-specific survival rate, and different curves represent the population survival rate at different ages. Due to individual differences of aphids, the population survival rate will gradually increase at a certain stage. When all aphids develop into adults, the population survival rate will tend to flatten out. With the growth and development of aphids, the population survival rate will gradually decline and finally reach to 0. Taking the age-specific survival rate of T. aurantii on ZK in Figure 3 as an example, adults appear in the population around the 4th day, and the population survival curve gradually increased with the growth of aphids. Around 8th day, all aphids in the population developed into adults, and the population survival curve leveled off. Around 10th day, some individuals in the population die, and the population survival curve gradually declines. The graph is based on the life table created by Professor Chi Hsin and the sigmaplot software. Here are the references:

  1. Remzi A, Ismail K, M. Salih, Evin P, and Chi H. Population Growth of Dysaphis pyri (Hemiptera: Aphididae) on Different Pear Cultivars with Discussion on Curve Fitting in Life Table Studies. Journal of Economic Entomology. 2017, 110, 1890-1898. [http://doi.org/10.1093/jee/tox174]

Point 5: Line 229 and 232. Figure. 2 and Figure. 3 should be Figure 2 and Figure 3.

Response 5: We were really sorry for our careless mistakes. Thanks for your reminder. The corrections list in line 233 and 236.

Point 6: Line 233. Remove “turnip”.

Response 6: According to the reviewer’s comment, this sentence has been revised. The corrections list in line 236.

Point 7: Line 248. [282830]?

Response 7: Thanks for your careful checks, we are sorry for our carelessness. Here is the link to the reference, we have modified on line 251.

Point 8: Line 253-254. Not clear. Rephrase it.

Response 8: We sincerely thank the reviewer for careful reading. I'm sorry that this sentence caused the reviewer to have doubts, and we have rephrased it again. The correction lists in line 256-257.

Point 9: Line 300. “oviposition” should be “larviposition”.

Response 9: As suggested by the reviewer, we have corrected the “oviposition” into “larviposition”.  The correction lists in line 303.

Reviewer 2 Report

Dear Author,

Thank you for submitting your article titled "Different Tea Germplasms Distinctly Influence the Adaptability of Toxoptera aurantii (Hemiptera: Aphididae)" for review. I have carefully reviewed the manuscript and would like to provide you with my evaluation and suggestions.

Overall, the article presents valuable research on the adaptability of Toxoptera aurantii on different tea germplasms. The findings reveal distinct feeding behavior and population dynamics of T. aurantii on various host plants, indicating the potential for resistance breeding and integrated pest management. The results are significant for the field of plant-aphid interactions and contribute to our understanding of aphid adaptation.

However, there are some areas that require improvement. Firstly, in the introduction section, the explanations need to be more concise. It is advisable to provide a brief and focused overview of the importance of understanding the feeding behavior and population dynamics of aphids for host adaptation and resistance screening.

Secondly, the language and grammar in the manuscript need revision. Some sentences are unclear or lack proper syntax. I recommend that you carefully proofread and edit the manuscript to ensure clarity and coherence throughout.

Regarding Figure 2, and Figure 4, I suggest providing additional explanations to enhance their comprehensibility. The captions should include concise descriptions of the data presented and the key findings or trends. Consider using labels or annotations within the figures to highlight important features or variables.

In conclusion, this article presents valuable research findings and contributes to the field of insect molecular biology and genomics, specifically the biology and molecular mechanisms of plant-aphid interactions. However, improvements are needed in the introduction section to provide a more focused context, and the language and grammar should be revised for clarity and coherence. Additionally, Figure 2 and Figure 4 require further explanations to improve their clarity and enhance the understanding of the data.

Once these revisions are made, the article will be suitable for publication. I encourage you to carefully address the points mentioned above and resubmit your manuscript for further consideration.

Thank you for your contribution to the field of insect biology, and I look forward to seeing the revised version of your article.

Best regards,

Dear Author,

Thank you for submitting your article titled "Different Tea Germplasms Distinctly Influence the Adaptability of Toxoptera aurantii (Hemiptera: Aphididae)" for review. I have carefully reviewed the manuscript and would like to provide you with my evaluation and suggestions.

Overall, the article presents valuable research on the adaptability of Toxoptera aurantii on different tea germplasms. The findings reveal distinct feeding behavior and population dynamics of T. aurantii on various host plants, indicating the potential for resistance breeding and integrated pest management. The results are significant for the field of plant-aphid interactions and contribute to our understanding of aphid adaptation.

However, there are some areas that require improvement. Firstly, in the introduction section, the explanations need to be more concise. It is advisable to provide a brief and focused overview of the importance of understanding the feeding behavior and population dynamics of aphids for host adaptation and resistance screening.

Secondly, the language and grammar in the manuscript need revision. Some sentences are unclear or lack proper syntax. I recommend that you carefully proofread and edit the manuscript to ensure clarity and coherence throughout.

Regarding Figure 2, and Figure 4, I suggest providing additional explanations to enhance their comprehensibility. The captions should include concise descriptions of the data presented and the key findings or trends. Consider using labels or annotations within the figures to highlight important features or variables.

In conclusion, this article presents valuable research findings and contributes to the field of insect molecular biology and genomics, specifically the biology and molecular mechanisms of plant-aphid interactions. However, improvements are needed in the introduction section to provide a more focused context, and the language and grammar should be revised for clarity and coherence. Additionally, Figure 2 and Figure 4 require further explanations to improve their clarity and enhance the understanding of the data.

Once these revisions are made, the article will be suitable for publication. I encourage you to carefully address the points mentioned above and resubmit your manuscript for further consideration.

Thank you for your contribution to the field of insect biology, and I look forward to seeing the revised version of your article.

Best regards,

Author Response

Dear Editors and Reviewers:

Thank you for your letter and for the reviewers’ comments concerning our manuscript (ID: insects-2462631). Those comments are all valuable and very helpful for revising and improving our paper, as well as the important guiding significance to our researches. The manuscript has been carefully revised, and the revisions were marked in red. The point-by-point answers to the comments and suggestions were listed below.

Response to Reviewer 2 Comments

Point 1: there are some areas that require improvement. Firstly, in the introduction section, the explanations need to be more concise.  It is advisable to provide a brief and focused overview of the importance of understanding the feeding behavior and population dynamics of aphids for host adaptation and resistance screening. Secondly, the language and grammar in the manuscript need revision.  Some sentences are unclear or lack proper syntax.  I recommend that you carefully proofread and edit the manuscript to ensure clarity and coherence throughout.

Response 1: Thanks for your suggestion. We have trid our best to polish the language in the revised manuscript. In addition, we have added an brief and focused overview about the importance of understanding the feeding behavior and population dynamics of aphids for host adaptation and resistance screening in the introduction according to your suggestion, and the relevant content is in lines 64 to 70 of the manuscript. We appreciate for Editors/Reviewers’ warm work eamestly and hope that the correction will meet with approval.

Point 2: Regarding Figure 2, and Figure 4, I suggest providing additional explanations to enhance their comprehensibility.  The captions should include concise descriptions of the data presented and the key findings or trends.  Consider using labels or annotations within the figures to highlight important features or variables.

Response 2: We think this is an excellent suggestion. We have already annotated the manuscript in lines 232-233 and 238-240.